# OpenReview forum: "Data Matters Most: Auditing Social Bias in Contrastive Vision–Language Models"
_TMLR — Accepted by TMLR_

### Review · Reviewer_8dvH · 2025-07-30

**Summary Of Contributions:**

This paper presents a contribution to the field of AI fairness by conducting a systematic and controlled audit of social bias in the scaling law of contrastive VLMs. The authors' primary contribution is the experimental framework itself, which successfully disentangles three critical factors of scaling law: model size, training data scale, and training data source. By holding the contrastive loss function constant and comparing CLIP and OpenCLIP models across these axes, the work provides clear, empirical evidence that challenges common intuitions about scaling up.

**Audience:**

Yes

**Claims And Evidence:**

Yes

**Requested Changes:**

Please see the last bullet point of weakness. I would request the authors to conduct a couple of bias mitigation methods and compare how they perform under scaling law.

**Strengths And Weaknesses:**

Strengths:
- The paper's primary strength is its controlled experimental framework that systematically assessed the impact of scaling law on social bias. I believe this perspective from scaling law is novel.
- The research provides compelling empirical evidence that refutes the common intuition that larger models or bigger datasets are inherently fairer. It demonstrates that naïve scaling can amplify, rather than mitigate, social biases. The conclusion could provide guidance on how to better scale up foundation models.
- The authors release all code and evaluation scripts, which enables transparent and reproducible auditing of other and future VLMs.

Weaknesses:
- The analysis is confined to binary gender and seven race categories, and does not evaluate finer-grained ethnicities or intersectional subgroups (e.g., young, Black, female). Additionally, the bias probes are limited to well known stereotypes, leaving other potential biases like political ideology or socioeconomic status unexplored.
- The study only focuses on two model families (CLIP and OpenCLIP) and two architectural sizes. The results may not generalize to newer multimodal architectures.
- All experiments use a single, canonical prompt structure ("a photo of a..."). The authors acknowledge that different prompt engineering strategies might alter the measured bias scores.
- It is unknown whether there are overlaps between the evaluation datasets (FairFace, PATA) and the large-scale pre-training corpora. The authors should check whether the evaluation datasets has already been contaminated.
- This is only an analysis of evaluation. I would expect the authors to also compare a couple of bias mitigation methods to see how they perform under the scaling law. For example, please check Algorithm 1 in Janghorbani & De Melo (2023).

> Sepehr Janghorbani and Gerard De Melo. Multi-modal bias: Introducing a framework for stereotypical bias
assessment beyond gender and race in vision–language models.

---

> ### Author Response · Authors · 2025-08-15
> **Response to Reviewer 8dvH**
>
> Thank you for your valuable feedback and the positive remarks about our paper. We have carefully considered your comments and made corresponding revisions to address the concerns raised. The full updated manuscript is also now uploaded, addressing all requested revisions.
>
> •   **Mitigation methods added**
>
> Per your request, we now compare three debiasing strategies under the same size/scale/source controls: Bias Prompts, Prompt Array, and SANER. As a summary: no universal winner—Bias Prompts work best for gender on CLIP-B/32; Prompt Array best addresses OpenCLIP race (crime/communion); SANER is uniquely consistent for race–agency; scaling LAION 400M→2B reorders best method.
>
> •	**Prompt robustness**
>
>  In addition to the canonical “photo of a …” template, we evaluate two prompt variants (App. C). Trends hold; Section 4 reports prompt-engineering debiasing effects explicitly.
>
> •	**Generalization scope**
>
> While we focus on CLIP/OpenCLIP and ViT-B/32/L/14, these encoders are widely used as vision backbones in recent MLLMs; our controlled design isolates size/scale/source effects that those systems inherit.
>
> •	**Demographic coverage**
>
>  We acknowledge the current scope (binary gender, seven race categories). We add discuss limitations of our approach to facilitate intersectional analyses (e.g., age×race×gender) in future research.
>
> •	**Data overlap**
>
>  As CLIP data is private, fairface was used for fairness analysis in the original CLIP paper and in prior audits. PATA  was released postdate the pretraining of CLIP and OpenCLIP.
>
>
> We hope these additions address your concerns: we compare debiasing methods under scaling, verify prompt/template stability, and discuss generalization limits.

---

### Review · Reviewer_hDoS · 2025-08-05

**Summary Of Contributions:**

This paper presents a systematic investigation that disentangles three key factors affecting bias in vision-language models: model size, training data scale, and training data source that challenges common assumptions about the effect of scaling on bias. Through controlled experiments comparing CLIP and OpenCLIP models while keeping the contrastive objective constant, the authors demonstrate that larger models and datasets can sometimes increase rather than reduce bias.  Their findings reveal that the training data source is the most important factor of bias patterns, with CLIP showing more gender fairness but racial bias, while OpenCLIP exhibits the opposite pattern, highlighting the need for data-centric mitigation strategies.

**Audience:**

Yes

**Broader Impact Concerns:**

None.

**Claims And Evidence:**

Yes

**Requested Changes:**

Proposed adjustments:
1. Add control experiments with neutral images to establish whether patterns represent social biases rather than embedding space artifacts.
2. Discuss whether biases persist or diminish when models undergo fine-tuning for practical applications, which is crucial for assessing real-world impact.
3.  Provide additional information on how the magnitude of embedding distance links to bias in practical terms (e.g. significance of values near zero).

Other:
- Investigate the effect of different template formats would better help understand if the biases are present consistently across different variations.

**Strengths And Weaknesses:**

Strengths
- Presents a methodologically rigorous approach that isolates three distinct factors affecting bias in vision-language models by keeping the contrastive loss function constant.
- Demonstrates that training data source is a more important factor than model size which provides actionable insights for VLM development.
- Performs social-perception bias audits on four common open-source models that should be of interest for the community.
- The authors' release of evaluation code and scripts promotes reproducibility and standardization in studying the effect of bias in multimodal embedding models.

Weaknesses
- Relies on embedding distances without focusing on control experiments using perceptually similar but demographically neutral images. Without these controls, it's difficult to determine whether the observed patterns truly represent social biases rather than general properties of the embedding space.
- Focuses on pre-training biases without exploring how downstream fine-tuning or classification layers, which are typically used in practice, might mitigate these representation biases.
- Evaluates models exclusively on facial analysis, which may not generalize to how bias manifests in other common vision-language tasks like image captioning, visual question answering, or scene understanding.

---

> ### Author Response · Authors · 2025-08-15
> **Response to Reviewer hDoS**
>
> Thank you for your valuable feedback and the positive remarks about our paper. We have carefully considered your comments and made corresponding revisions to address the concerns raised. The full updated manuscript is now uploaded, addressing all requested revisions.
>
> •	**Neutral-image controls → Addressing “embedding artifacts”**
>
> We added §Neutral-Image Control in Appendix: 19 feature-neutral portraits show tight clustering and near-zero bias. Directional bias collapses to |Δ|≤0.02 for COMM/AGEN and ≤0.05 for CRIME, an order of magnitude below FairFace/PATA skews—confirming our metrics aren’t spuriously triggered by embedding geometry alone.
>
> •	**Calibration of bias magnitude → Linking Δ to practical harm**
>
> We added §Calibration of Directional-Bias Magnitude in Appendix: per-model calibration curves map |Δ| to the probability of a harmful label. For CLIP-B/32, risk crosses 50% near |Δ|≈0.25; for OpenCLIP, the elbow is earlier (0.1–0.2) and saturates ≥95% by |Δ|=0.3. This provides action thresholds (moderate vs. severe) and grounds Δ in end-user risk.
>
> •	**Template robustness → Prompt format variations**
>
> We added §Robustness to Prompt-Template Variation with two alternatives (“an image of…”, “portrait of…”) in the Appendix. Across all models/tasks, the mean shift is Δμ ≤ 0.02 cosine units and the rank order is unchanged. Patterns are template-stable.
>
> •	**Fine-tuning/mitigation → Do biases persist under adaptation?**
>
> While full backbone fine-tuning is out of scope, we added three mitigation strategies that represent common practical adaptations: Bias Prompts (training-free projection/calibration), Prompt Array (adversarial learned tokens), and SANER (lightweight neutralizer layer). Results across size/scale/source: no universal winner; Bias Prompts best for gender on CLIP-B/32, Prompt Array best for OpenCLIP race (crime/communion), SANER uniquely consistent for race–agency; scaling 400M→2B reorders best method. These findings indicate partial reductions but persistent biases, motivating future work on task-tuned adapters/backbone FT (discussed in the updated limitations).
>
> •	**Task scope beyond faces → Generalization**
>
> We motivate faces for balanced labels, but discuss implications for open-vocabulary VLM backbones and flag this as future work in Limitations/Discussion.
>
> We hope these additions address your concerns: we verify the metrics are not geometric artifacts, tie magnitudes to harm probability, examine practical debiasers, and clarify generalization limits and next steps.

---

> > ### Comment · Reviewer_hDoS · 2025-09-04
> > **Response to authors**
> >
> > Thank you for thoroughly addressing my requests with substantial additional analysis.
> >
> > The neutral control experiment establishes an important baseline, showing the metrics aren't triggered by embedding geometry alone. The calibration curves, template robustness tests, and mitigation strategy comparisons add critical context and practical utility to the work.
> >
> > These additions satisfy my concerns and significantly strengthen the paper.

---

### Review · Reviewer_Fdug · 2025-08-14

**Summary Of Contributions:**

This paper investigates how three factors (model size, training-data scale, and training-data source) shape social bias in contrastive vision–language models like CLIP and OpenCLIP. They use a controlled experimental design that keeps the loss function constant. The authors compare seven model–data combinations across balanced face-analysis benchmarks (FairFace and PATA) with bias probes for criminal/animal associations and social stereotypes (communion and agency).

They find that scaling effects are neither uniform nor universally beneficial: enlarging CLIP’s encoder reduces gender bias without affecting race bias, but the same change in OpenCLIP increases both; expanding LAION data from 400M to 2B pairs consistently worsens bias; and at matched budgets, CLIP’s proprietary data is more race-biased while LAION data is more gender-biased. Race-related skew is generally higher than gender skew, and harmful labels like negative communion occur far more often than crime mislabelling.

**Audience:**

Yes

**Broader Impact Concerns:**

This paper uses CLIP's proprietary dataset, which lacks basic descriptions and statistics. It becomes hard to assess the broader impact of using this dataset.

**Claims And Evidence:**

Yes

**Requested Changes:**

- There are many sentences missing a space at the beginning of the sentence. Please fix similar typos in the draft.
- It would be better to first provide a summary of notation definitions before presenting the metrics.
- Please provide a more detailed explanation for the considered metrics in this paper.
- For other changes, please see the weaknesses

**Strengths And Weaknesses:**

### Strengths

- This paper proposes to disentangle model size, data scale, and data source while holding the loss function constant, allowing clear attribution of bias changes to specific factors.
- This paper uses two balanced benchmarks (FairFace, PATA) and diverse bias probes, and releases code and scripts. The experimental results challenge common “bigger is fairer” beliefs.

### Weaknesses
- The reason why these two datasets lead to different biases is not thoroughly explored. For example, what are the distribution differences between the two datasets? How are the distributions of the datasets related to the test benchmark? Additionally, how do the authors access CLIP's proprietary data?
- The model sizes studied in this paper are limited. This paper only considers two model sizes. Do the results hold for a broader range of sizes? The same applies to data set sizes. More fine-grained studies are necessary.
- Are the biases originating from the language source or the image source? It would be interesting to further explore.
- How do the conclusions from the studies guide the future training of vision-language models?

---

> ### Author Response · Authors · 2025-08-15
> **Response to Reviewer Fdug**
>
> Thank you for the thoughtful review and positive remarks. We’ve revised the manuscript; the updated version addresses all points.
>
> •	**Datasets & why they differ**
>
> We expanded §3.2 Benchmarks to highlight that PATA is a context-based bias benchmark (scenes + positive/negative captions), whereas FairFace contains cropped face-only images (“face shots”) with no scene context—explaining why the datasets stress different bias patterns.
>
> •	**Access to CLIP data. We do not access CLIP’s private corpus**
>
>  We evaluate released CLIP checkpoints (≈400M pairs) under a fixed loss, holding model size and objective constant to isolate data source effects; we acknowledge the limits this imposes on auditing the proprietary crawl (see Limitations).
>
> •	**Size/scale coverage**
>
>  We study the widely used ViT-B/32 and ViT-L/14 and LAION 400M/2B because they are the most adopted in practice; we added a Limitations subsection outlining plans for finer-grained size/scale sweeps and newer backbones.
>
> •	**Image vs. text source of bias**
>
> Prior work suggests vision often amplifies leakage [1]; in our controls, the text encoder is shared/frozen across conditions, and we vary vision-encoder size + data source/scale. We added neutral-image controls (App. A) showing near-zero directional bias (|Δ|≤0.02 for COMM/AGEN; ≤0.05 for CRIME) and calibration curves (App. B) mapping |Δ| to harm probability—evidence that the measured effects are not embedding artifacts and that risk rises steeply in OpenCLIP.
>
> •	**Guidance for future training**
>
>  We now make explicit recommendations: prioritize data-centric interventions (rebalancing/augmentation), monitor both disparity (Max-Skew) and harm frequency, select mitigation per model family/axis, and validate at the deployment size (scaling is not uniformly fair). We also report three test-time debiasers (Bias Prompts, Prompt Array, SANER) under identical controls—no universal winner; efficacy depends on source and size.
>
> •	**Requested edits**
>
>  We fixed spacing/typos; added a notation summary before the metrics; and expanded metric definitions (Max Skew, Harm Rate) with formulas and a worked example.
>
> •	**Broader impact of using CLIP’s dataset**
>
>  We explicitly state that the proprietary corpus lacks public statistics; we therefore (i) compare against LAION at matched budgets, (ii) include harm frequencies alongside disparities, and (iii) soften claims where source auditability is limited (see Limitations).
>
> We hope these revisions address your concerns and make our contributions—and their scope—clear.
>
> [1] M. Hall, L. Gustafson, A. Adcock, I. Misra and C. Ross, "Vision-Language Models Performing Zero-Shot Tasks Exhibit Disparities Between Gender Groups," 2023 IEEE/CVF International Conference on Computer Vision Workshops (ICCVW), Paris, France, 2023, pp. 2770-2777, doi: 10.1109/ICCVW60793.2023.00294.

---

> > ### Comment · Reviewer_Fdug · 2025-09-01
> > **Thanks to the authors for the revision**
> >
> > Thanks to the authors for their revision to address my previous concerns. My concerns have been addressed. Therefore, I have adjusted the recommendations for this paper.

---

### Decision · Action_Editor_basD · 2025-10-03

**Recommendation:** Accept as is

**Audience:**

Yes

**Audience Explanation:**

The subset of the community interested in VLMs and interested in social bias in models

**Claims And Evidence:**

Yes

**Claims Explanation:**

Th reviewers seem to agree that the claims are supported by empirical evidence in this paper